Optimizing the amount of pig manure in the vermicomposting of spent mushroom (Lentinula) substrate

Shi Yajing
Wang Zhenyu wzy150618@sina.com
Wang Yurong
School of Biological and Chemical Engineering, Liaoning Institute of Science and Technology , Benxi , Liaoning , China
Song Young-Chae
Electronic publication date: 2020 Dec 18
Publication date: 2020
Volume: 8
Electronic Location ID: e10584
Received 2020 Aug 28; Accepted 2020 Nov 24
Copyright: ©2020 Shi et al.
Copyright year: 2020
Copyright holder: Shi et al.
License: This is an open access article distributed under the terms of the Creative Commons Attribution License, which permits unrestricted use, distribution, reproduction and adaptation in any medium and for any purpose provided that it is properly attributed. For attribution, the original author(s), title, publication source (PeerJ) and either DOI or URL of the article must be cited.
License URL: https://creativecommons.org/licenses/by/4.0/

Keywords: Spent mushroom substrate (SMS), Pig manure addition, Vermicomposting, Optimizing

Funding: National Natural Science Foundation of China 41701300 Scientific Research Foundation of Liaoning Province 201601338 Basic Research Project of Colleges and Universities in Liaoning Province LJ2017QL015 This work was funded by the National Natural Science Foundation of China [grant number 41701300], the Scientific Research Foundation of Liaoning Province [grant number 201601338], and the Basic Research Project of Colleges and Universities in Liaoning Province [grant number LJ2017QL015]. The funders had no role in study design, data collection and analysis, decision to publish, or preparation of the manuscript.

==============================
Background

The mushroom industry produces a large amount of spent mushroom substrate (SMS), which requires a large geographical footprint and causes pollution.

Methods

We sought to optimize the C:N ratio of the initial feedstock used in vermicomposting of SMS by adding pig manure additions. We applied five treatments to the initial feedstock (S0, S1, S2, S3, and S4) with different C:N ratio of approximately 35, 30, 25, 20, and 15, respectively.

Results

Our results showed that lignin and cellulose in SMS were degraded after 56 days vermicomposting, especially in S2 (77.05% and 45.29%, respectively) and S3 (65.05% and 48.37%, respectively) treatments. We observed the degradation of the fibrous structure in SMS using pig manure treatments after vermicomposting by microscope and scanning electron microscope. Cellulase and polyphenol oxidase (PPO) were enhanced in pig manure treatments during vermicomposting, especially in the S2 and S3 treatments. The biomass of earthworms in the S2 treatments was at its highest level among all treatments at 28 to 56 days. The high level of PPO activity in the S2 treatment may protect cellulase and earthworms against the aromatic toxicity that is a byproduct of lignin degradation, particularly at 28 to 56 days of vermicomposting. Conclusively, it indicated that the C/N ratio of 25 in the S2 treatment was the optimal for SMS vermicomposting with the addition of pig manure. Our results provide a positive application for the recycling of both SMS and pig manure.

Introduction

Global mushroom production has increased more than 30 fold since 1978, from about 106 metric tons in 1978 to 3.4 × 107 metric tons in 2013 (Daniel, Johan & Qi, 2017). China is a major mushroom producer, accounting for 87 percent of global production in 2013 (Daniel, Johan & Qi, 2017). Mushroom output reached at 3.78 × 107 metric tons in China in 2018 alone (China Edible Fungi Association, 2019). One kg harvest of mushrooms will result in 5 kg of spent mushroom substrates (SMS) (Jordan, Mullen & Murphy, 2008). For example, the output of Lentinula reached 1.04 × 107 metric tons in 2018 (China Edible Fungi Association, 2019), which resulted in over 5.2 × 107 metric tons of SMS in 2018. Untreated SMS may cause environmental problems, and the recycling or disposal of SMS is a crucial issue, especially in China.

Traditionally, SMS is disposed of by directly returning the waste to the field and composting, which has some practical disadvantages. (1) SMS directly returned to the croplands may cause nitrogen competition between the crops and the microbes in the soil due to the high carbon content of SMS. It may harm crop growth, at least in the short term. (2) If SMS was processed by compost, the compost pile typically contains at least nearly a ton of agricultural waste, which occupies a large footprint on land. The composting process itself often requires professional technical support to turn the compost, and lasts for 90 to 120 days phase of treatment (Busato et al., 2012). These considerations make it difficult for Chinese farmers and other mushroom producers to deal with SMS. However, vermicomposting can be piled any size and can be composted in 21 to 60 days (Suthar & Singh, 2008). It requires minimal materials and does not need complicated skills such as the turning and sealing required for in composting. Maintaining vermicomposting only requires adding water to the pile when necessary. Thus, vermicomposting may be an efficient way to dispose of SMS.

Moreover, the product of SMS vermicomposting could replace a portion of inorganic fertilizers (Jat & Ahlawat, 2006). A study showed that the germination rates of maritime pine were increased by 16% after the addition of vermicompost into growing media (Lazcano, Sampedro & Zas, 2010). Germination rates and early growth of wheat were also enhanced by vermicompost amendment (Hussain, Abbasi & Abbasi, 2016). The growth of shoots, roots, and leaves in the seedlings of tomato and French marigold were improved by the addition of vermicompost (Bachman & Metzger, 2008). Moreover, the C, N, P, K, Ca, Mg, L-ascorbic acid, glucose, and fructose content in tomato fruits were also increased with the addition of vermicompost (Zaller, 2007).

The vermicomposting is a process accomplished by the interaction of earthworms and microbes. The C:N ratio of the initial feedstock is vital in vermicomposting and influences the growth and activity of microbes and earthworms (Aira, Monroy & Domínguez, 2006; Biruntha et al., 2020). Firstly, a high C:N ratio treatment slows microbial decomposition by limiting nitrogen (Eiland et al., 2001). A low C:N ratio treatment decreases the microbes activity by exhausting carbon nutrients (Eiland et al., 2001). Kumar et al. (2012) reported that C:N ratios from 13.9 to 19.6 were the optimal for composting. Secondly, the C:N ratio of the initial feedstock influences on the living condition of earthworms as well. Schönholzer et al. (1998) reported that the palatability of different leaves to earthworms was dependent upon their C:N ratio. Empirical results showed that the best vermicomposting would have a C:N ratio of 25 (Ndegwa & Thompson, 2000) and successful vermicomposting is dependent on the correct. Therefore, it is crucial to optimize N-rich substances when adjusting the initial feedstock with C-rich SMS for successful vermicomposting.

Animal manure contains high levels of nitrogen. For example, the values of C:N ratios in the manure of layer hens, pigs, and beef cattle are approximately 10, 13, and 25, respectively. The global output of beef cattle reached 296 million in 2018. In China, approximately 1,105 million layer-hens, 396 million pigs, and 44 million beef cattle were slaughtered in 2018. The production of poultry and livestock creates a vast amount of animal manure that must be processed (Rehman et al., 2017). Each layer hen, pig, and beef cattle is estimated to produce about 34 kg, 2,000 kg, and 11,700 kg of manure annually, respectively (United States Government Accountability Office, 2008). Of these sources, pig manure with the higher N content and the largest amount, is most suitable for use in vermicomposting SMS.

Little research has been conducted on vermicomposting SMS with pig manure. There are even fewer studies focused on the physical, chemical, and biological changes of the materials during vermicomposting. He et al. (2017) used Fourier transform infrared (FTIR) spectroscopy, scanning electron microscopy (SEM) and Brunauer–Emmett–Teller (BET) to determine the structural changes of the vermicompost product. Gong et al. (2019) reported on the chemical and biological changes of SMS, such as the biomass of earthworms, and the contents of cellulose and lignin during vermicomposting. These changes directly reflect the degradation of vermicompost materials. We conducted a single factor study of vermicomposting to dispose of SMS with the C:N ratio as the variable.

We evaluated the optimal proportion of pig manure addition in SMS vermicomposting based on its physical, chemical, and biological parameters. We used the paraffin method and scanning electron microscopy (SEM) detection to assess the physical properties of SMS. The lignocellulose content and EC values were measured to evaluate the chemical properties of SMS. The biomass of earthworm and enzyme were used to analyze the biological changes of SMS. These measurements contributed to understanding the mechanism of the SMS vermicomposting process, especially the decomposition of lignocellulose.

Materials & Methods

Experimental design

SMS was collected from a mushroom farm in Benxi, China near the Liaoning Institution of Science and Technology. The medium used in shiitake production is generally composed of 49% wood chips, 30% cottonseed shell, 15% bran, 5% corn flour, and 1% gypsum. The cellulose and lignin content in SMS were 32.32% and 6.74%, respectively. Pig manure was collected from a pig farm near the Liaoning Institution of Science and Technology. All nonbiological material such as plastic, stone, and glass were removed manually from the SMS before the experiment.

We used five treatments with different C:N ratios in the initial feedstock. The C content of SMS and pig manure was 193.2 and 259.4 g kg−1, respectively, and the N content of SMS and pig manure were 68.1 and 202.5 g kg−1, respectively. The amounts of SMS and pig manure were calculated according to the different C:N ratios. The C and N content were measured following the procedure in the physicochemical characteristics. In details, SMS and pig manure were mixed with different C:N ratios of approximately 35, 30, 25, 20, and 15 in S0 (SMS 300 g), S1(SMS 297 g and pig manure 3 g), S2 (SMS 270 g and pig manure 30 g), S3 (SMS 225 g and pig manure 75 g), and S4 (SMS 180 g and pig manure 120 g), respectively. Three hundred grams of initial feedstock (dry weight) were placed in a 1 L glass beaker and 200 mL water was added and left to sit for one week. Earthworms were handpicked, washed, and placed in a container with wet filter paper overnight to let the worms empty their gut contents. We chose adult worms with a reproductive ring with individual weights between 0.35 and 0.45 g. The biomass of the earthworms was the weight of worms with empty guts. Earthworms were added at 5 pieces per 100 g feedstock. Thirty earthworms (Eisenia fetida) were placed in each container into S0, S1, S2, S3, and S4 treatments, respectively. There were four replicates in each treatment and total 100 microcosms (A 1 L glass beaker, covered with two layers of gauze to allow for aeration, used for destructive sampling). All of the microcosms with the earthworm and SMS mixture were cultured in an artificial climate chamber at 25 °C with 60–70% humidity. Vermicomposting lasted for 56 days and the moisture content was maintained at 60 ± 10% using distilled water once a week in proportion to the weighted loss of moisture. No other feed materials were added during the 56 days of vermicomposting. Subsamples from the different treatments were collected every two weeks. At the sampling day, earthworms were manually handpicked on each sample day and the indicators of SMS were measured.

Physical characteristics of SMS vermicomposting

We collected subsamples from each treatment on day 56 of vermicomposting. The subsamples were dried at 45 °C to a constant weight using drying oven.

After drying, the subsample in each treatment was embedded in melted paraffin at 50−60 °C and solidified at room temperature. The subsample was immobilized by overnight placement. The subsample was sectioned with a paraffin slicer. Sections were supported on glass slides coated with an adhesive (a mixture of egg whites and glycerin) and placed in a water bath at 45 °C. The sections were dewaxed and Canada gum were added for observation. We used a microscope to obtain microscopic images of SMS.

The subsample in each treatment was placed on conductive tape and cleared subsamples without adhesion. After spraying, the subsamples were observed with SEM (Thermo Fisher, Quanta 250).

Physicochemical characteristics of SMS vermicomposting

The total organic carbon content was determined using potassium dichromate oxidation –spectrophotometry (Nelson & Sommers, 1996). Total nitrogen (TN) was determined using the Kjeldahl method (Jackson, 1973). The pH value and electrical conductivity (EC) were measured using the potentiometric method.

The cellulose and lignin contents were estimated using the Van Soest method (Van Soest, Robertson & Lewis, 1991) with some modifications. A five-gram sample was heated to boiling in 100 ml of neutral detergent (Sodium Lauryl Sulphate 30 g L−1, Ethylene Diamine Tetraacetic Acid (EDTA) 18.6 g L−1, boric acid 6.8 g L−1, Na2HPO4 4.56 g L−1, and diethyl ether 10 mL L−1) plus 50 µ1 of heat stable amylase to remove any starch. The sample were heated in ethanol and filtered to remove any lipids; H2SO4 was added (1N) Cetrimonium Bromide (CTAB, 20 g L−1) to remove any protein. The pretreated sample was dried and 0.1 g of the pretreated sample was added to 10 mL acetic acid (10 g L−1) to determine the amount of cellulose. Alternatively, the pretreated sample was added to 10 mL acetic acid plus nitric acid (1:1, volume by volume), shaken for 5 min, centrifuged for 10 min at 8000 g to collect andy deposits, and then three mL H2SO4 (12 mol L−1) was added and left overnight to completely dissolve any cellulose. 10 mL water was then added, stirred, and heated to boiling for 10 min; 0.5 mL BaCl2 (100 g L−1) was added, stirred, and centrifuged for 10 min at 8,000 g to collect deposits. 10 mL potassium dichromate sulfate solution (0.5 N) was added to this and heated to boiling with constant stirring. The remaining deposits were washed with 10 mL water. Mohr’s salt solution titration was used to determine the cellulose content and iodometry was used to measure the lignin content in SMS.

Biological characteristics of SMS vermicomposting

The biomass, survival rate and mortality rate of earthworm

Earthworms were collected from all treatments by hand every two weeks. They were washed, allowed to dry on filter paper, and weighed. The biomass, survival rate, and mortality rate of the earthworms were recorded.

The activities of cellulase and polyphenol oxidase (PPO)

Cellulase activity was analyzed according to the method of Borgna (2004) and the reducing sugar generated from carboxymethyl cellulose sodium salt we quantified with some modifications. The reducing sugar content was determined using a method of 3,5-dinitrosalicylic acid (DNS). Five grams of sample were diluted with 0.1 M Sodium acetate buffer (pH 5.5) to 100 mL and incubated at 4 °C for 24 h. After 10 min of equilibrium, two mL diluent mixed with five mL DNS solution (3.15 g L−1 DNS, 91 g L−1 sodium tartrate, 2.5 g L−1 phenol, and 2.5 g L−1 sodium sulfite), and two mL carboxymethyl cellulose sodium salt (0.8 g L−1) was added and incubated for 30 min under 37 °C. The rate of change was determined spectrophotometrically at 540 nm. One unit (U) of cellulase activity is the production of 1µg mL−1 reducing sugar from 4 mg mL−1 carboxymethyl cellulose sodium salt amount within one minute under 37 °C.

The PPO activity was determined according to the method of Gawlik-Dziki, Złotek & Świeca (2008) with modification. A 5 g sample was mixed with 100 mM catechol and 0.1 M K-phosphate buffer (pH 6.88), and incubated for 10 min under 40 °C. Then added two mL trichloroacetic acid (20%) was added to stop the enzymatic reaction and the rate of change was determined spectrophotometrically at 410 nm. One unit (U) of enzyme activity was calculated as the amount of enzyme required for an increase of 0.001 in the absorbance value in one minute.

Seed germination, root and shoot growth in seedling time

Aqueous extracts were prepared from five treatments with distilled water (1:80 w/v). The phytotoxicity of these extracts was evaluated through a seed germination test according to the procedures of Abdullahi et al. (2008) with some modifications. A seed germination bioassay for maize was evaluated in which the Whatman filter paper was placed inside a sterilized petri dish and wetted with extracts. 25 corn seeds were placed on top of filter paper and incubated for five days in the dark at 25 °C. The experimental control used filter paper wetted with sterilized water. Seed germination, the percentage of relative seed germination (RSG, relative root elongation (RRG), and the germination index (GI) were calculated (Tiquia & Tam, 1998).

The results were reported as the mean of four replicates with standard error (SE). One-way ANOVA with Tukey’s HSD test was used to test the effect of different pig manure additions on the degradation rates of the fibrous structure in SMS, the chemical properties of the vermicompost products, earthworm biomass, and enzyme activities, and seed germination. The general linear-regression analysis was used to investigate the relationships in the C:N ratio, the contents of lignin and cellulose, the activities of cellulase and PPO, and the biomass of earthworms. The probability level used for statistical significance was p < 0.05 for all tests. SPSS 18.0 (SPSS Inc., Chicago, IL, USA) was used for statistical analyses.

Results

Physical characteristics of SMS vermicomposting

The microscopic images of the SMS paraffin section before and after vermicomposting are shown in Fig. 1. After 56 days of vermicomposting, the fibrous structure of SMS was affected by the C/N ratio in the initial vermicomposting feedstock, especially in the degree of fragmentation in SMS. The paraffin section presented with changes in the fibrous structures in SMS. A number of fibrous structures are seen in slides A, B, and F, and there were less fibrous structures in C and D. The decrease of fibrous structures demonstrates the degradation of lignocellulose. The degree of fragmentation in SMS increased from 15 to 20 of the C:N ratios and decreased from 25 to 35 of the C:N ratios. The degree of fragmentation was similar between the S2 and S3 treatments. There were more fragments in the S2 and S3 treatments compared with the others.

Figure 1 Optical microscope images of SMS in different treatments after 56 days of vermicomposting (600× magnify).

The image A presented the physical characteristics of SMS before vermicomposting (A) under microscope, and B, C, D, E and F presented the physical characteristics of SMS in S0, S1, S2, S3 and S4 treatments after 56 days of vermicomposting under microscope. The arrow shows the fibrous structure of SMS under microscope. The intact and broken fiber structure represents the non-degraded and degraded fiber structure in SMS.

The SEM images of SMS before and after vermicomposting are shown in Fig. 2. The SEM images show the bright and smooth areas as unbroken fibrous structures in SMS. The fragmented areas show the degraded fibrous structures of SMS. The dark area in the middle of the figure reveals the holes of SMS (such as the porous cob of corn). The SMS surface was relatively smooth before vermicomposting with occasional fragments (Fig. 2A). After vermicomposting, the surfaces of SMS became rough, cracked, porous, and fragmented. The degradation rate of SMS increased with the addition of pig manure among the S0, S1, and S2 treatments (Fig. 2). The degradation rates of SMS between S2 and S3 were similar. However, further increasing of pig manure addition weakened degradation rates of SMS in S4 treatment.

Figure 2 Scanning electron micrographs (SEM) of SMS in different treatments after 56 days of vermicomposting.

The image A presented the physical characteristics of SMS before vermicomposting (A) under SEM, and B, C, D, E and F presented the physical characteristics of SMS in S0, S1, S2, S3 and S4 treatments after 56 days of vermicomposting under SEM. The arrow shows the fibrous structure of SMS under SEM. The smooth and fragmentation structures indicated undegraded and degraded parts in SMS.

Chemical characteristics of SMS vermicomposting

The values of EC fluctuated in all the treatments during the 56 days cultivation period (Table 1). The EC values in the treatments increased with cultivation with the exception of the S1 treatment. The EC values were 578 µm cm−1 and 615 µm cm−1 in the S3 and S4 treatments, respectively. The EC values were 64.48% and 55.47% higher in the S3 and S4 treatments than the control, respectively.

TOC and TN decreased in all treatments (Table 1). The vermicompost system led to a decreasing in nutrients during cultivation. The TOC content in the vermicompost treated with pig manure was similar in value and was higher than that without pig manure. The TN results had a similar tendency to TOC. The nutrients in the SMS mixtures rapidly degraded during the first 28 days and then slowed their consumption between days 28 to 56. The invariant of TOC and TN showed a similar tendency with the nutrient consumption in the vermicompost system. A slower nutrient consumption signaled the end of the vermicomposting.

Lignin and cellulose degradation increased with the vermicomposting process was affected by the addition of pig manure (Figs. 3A and 3B). The degradation rates of cellulose were 22.44%, 31.36%, 45.29%, 48.37%, and 21.61% in S0, S1, S2, S3, and S4 treatments, respectively, on the 56th day (Fig. 3C). The degradation rates of lignin were 26.86%, 58.78%, 77.05%, 59.05%, and 53.63% in the S0, S1, S2, S3, and S4 treatments, respectively (Fig. 3D). The degradation rates of cellulose were similar between the S2 and S3 treatment, but the degradation rates of lignin were 18.0% higher in the S2 treatment than the S3 treatment.

Table 1 Physico-chemical characteristics of the various substrates used for vermicomposting.

Values are mean ±  standard error (n = 3). Means in a row followed by different letters are significantly different at p < 0.05 according to Tukey HSD test.

Parameters	S0	S1	S2	S3	S4	One-way ANOVA	
						F value	P value	
Initial TN (g kg−1)	4.71 ±  0.29d	5.37 ±  0.14dc	6.13 ±  0.28c	7.54 ±  0.29b	10.19 ±  0.29a	64.26	0.000	
Final TN (g kg−1)	3.64 ±  0.37d	3.59 ±  0.16d	4.67 ±  0.45c	5.77 ±  0.21b	7.05 ±  0.21a	75.98	0.000	
Initial TOC (g kg−1)	26.32 ±  0.23a	25.02 ±  0.19b	24.91 ±  0.29c	23.76 ±  0.10d	22.92 ±  0.35e	10.51	0.001	
Final TOC (g kg−1)	23.27 ±  0.35a	20.51 ±  0.15b	20.01 ±  0.79b	20.09 ±  0.22b	20.12 ±  0.24b	5.36	0.014	
Initial pH	8.81 ±  0.03a	8.78 ±  0.04a	8.74 ±  0.03a	8.81 ±  0.03a	8.83 ±  0.03a	39.35	0.000	
Final pH	8.38 ±  0.14a	8.14 ±  0.13ab	8.01 ±  0.06b	7.99 ±  0.09b	7.60 ±  0.12c	18.11	0.000	
Initial EC (µs cm−1)	217.00 ± 9.54b	202.17 ±  8.28c	201 ±  7.81c	218.67 ±  8.02b	241.00 ±  6.56a	12.01	0.001	
Final EC (µs cm−1)	243.67 ±  14.22c	306.67 ±  12.74c	395.67 ±  44.66b	578.33 ±  81.09a	615.67 ±  13.20a	44.28	0.000	

Figure 3 Contents of cellulose and lignin; degradation rate of cellulose and lignin; and the activities of cellulase and PPO in different treatments during 56 days of vermicomposting.

Contents of cellulose (A) and lignin (B); degradation rate of cellulose (C) and lignin (D); and the activities of cellulase (E) and PPO (F) in different treatments during 56 days of vermicomposting. Vertical bars in the figures represent standard error of the means (n = 3).

Biological characteristics of SMS vermicomposting

(1) The earthworm biomass increased before 28 days and then decreased at 56 days (Table 2). The earthworm biomass increased with the addition of more pig manure in the S0, S1, and S2 treatments. The highest earthworm biomass was on day 28 in the S2 treatment among all treatments. Earthworm mortality was higher in the S0 treatment than other treatments at the end of the vermicomposting process (One-way ANOVA, F = 81.36, p < 0.05). Earthworm mortality was over 90% in the S0 treatment but was more than 73% in the S4 treatment. Earthworm mortality was approximately 25% in the S1 and S3 treatments. There was only one earthworm dead in one replicate of the S2 treatment during cultivation. The earthworm mortality decreased with the increase of pig manure addition among the S0, S1, and S2 treatments. However, further increasing of pig manure enhanced the earthworm mortality in S3 and S4 treatments.

Table 2 Biomass and mortality rates of the earthworm (Eisenia fetida).

Values are mean ±  standard error (n = 3). Means in a row followed by different letters are significantly different at p <0.05 according to Tukey HSD test.

Parameters	S0	S1	S2	S3	S4	One-way ANOVA	
						F value	P value	
Mean weight of earthworm in the initial time (g)	14.03 ±  5.5a	14.79 ±  8.1a	14.96 ±  14.8a	14.54 ±  3.0a	14.50 ±  16.7a	1.095	0.405	
Mean weight of earthworm at 28th day (g)	7.05 ±  1.05d	11.5 ±  1.45cd	17.6 ±  1.22a	13.07 ±  1.46b	9.70 ±  1.03c	29.69	<0.001	
Mean weight of earthworm at 56th day (g)	1.13 ±  0.99c	5.93 ±  1.16b	10.56 ±  1.03a	6.01 ±  0.67b	2.27 ±  0.31c	52.81	<0.001	
Mortality rate after 28 days (%)	36.67 ±  5.77a	20.00 ±  6.55b	0	6.67 ±  1.55b	23.33 ±  5.77ab	10.39	<0.001	
Mortality rate at 56 days (%)	90.00 ±  8.82a	24.44 ±  11.7c	1.11 ±  1.92d	26.67 ±  5.77c	73.33 ±  3.33b	81.36	<0.001	

(2) Figure 4 shows the results of the seed germination test with a 1.25% extraction of vermicompost products in all treatments. The Germination Index (GI) in all treatments was greater than 80%. The GI in the S2 treatment showed 36.09%, 10.29%, and 19.75% higher than that of the S0, S3, and S4 treatment, respectively.

Figure 4 Germination Indexes (GI) of Zea mays linn Spp (%).

A total of 1.25% extractions of SMS products in different treatments after 56 days of vermicomposting were used in this germination test. Vertical bars in the figures represent standard error of the means (n = 3).

(3) The cellulase activity was affected by different levels of pig manure addition (Fig. 3E). At 42 days, the cellulase activities in the S3 treatment reached its peak value (45.02 ± 0.04 mg glucose g−1 min−1) and was higher than that of the other treatments. At 56 days, the activities of cellulase showed higher in the S2 and S3 treatments than other treatments, but there were differences between the S2 and S3 treatments. The cellulase activities were lower in the S4 treatment than that of S0 and S1 treatments.

(4) PPO activity was increased by the addition of pig manure (except in the S4 treatment at 56 days) (Fig. 3F). At 56 days, an increased amount of pig manure decreased the PPO activity among pig manure addition treatments. The PPO activities ranged from greatest to least as 72.05 ± 6.93, 64.67 ± 4.37, 62.01 ± 6.12, and 25.33 ± 4.62 U g−1 min−1 in the S2, S1, S3, and S4 treatments, respectively.

Discussion

The high content of lignocellulose in SMS limited its degradation, which is an index to evaluate the efficiency of vermicomposting. We evaluated the changes in the physical, chemical, and biological properties in SMS to assess the effect of adding different amounts of pig manure for successful lignocellulose decomposing.

Changes of physical characteristics in SMS vermicomposting

Vermicomposting changed the structure of the SMS feedstock. Microscopic images showed the fiber structure in SMS was reduced by vermicomposting (Fig. 1) that the change was affected by adding different amounts of pig manure.

The fiber structure was most reduced in the S2 and S3 treatments when compared with the initial SMS. These results indicated that the C:N ratios of the initial feedstock at 20 and 25 was beneficial for SMS vermicomposting. Secondly, SEM images showed that SMS was destroyed and degraded by vermicomposting (Fig. 2). This explained the reduction of fiber structures seen in microscopic images of SMS. SEM images in this study were in agreement with those from previous studies (Gong et al., 2019; He et al., 2017; Soobhany, Mohee & Garg, 2015). The more fragments were seen in SMS implied a greater amount of degradation by vermicomposting. Fragmented SMS was beneficial to earthworm feeding and microbial contact. The fragmentation of the vermicomposting product is helpful for providing nutrients to plants. Thus, the C:N ratios of 20 and 25 in the initial feedstock were best for successful SMS vermicomposting.

Changes of chemical characteristics in SMS vermicomposting

Lignocellulose includes cellulose, lignin, and hemicellulose, which are the major structural components of plant material. The SMS used in our study contained 32.32% cellulose and 6.74% lignin (see results) and lignocellulose was the most prevalent fibrous structure in SMS. The recalcitrant lignin protects cellulose from degradation (Schmidt, Rye & Gurnagul, 1995). The degradation of lignocellulose determined the degradation of the fibrous structures in SMS. Vermicomposting induced fragmentation and degradation of the fiber structure in SMS (Fig. 1), which was similar to the results of Gong et al. (2019).

Vermicomposting is a process involving collaboration between earthworms and microbes. The SMS feedstock properties influence the feeding of the worm and the resulting microorganisms. The degradation of lignocellulose was affected by the dietary preferences and lifestyle habits of earthworms. The degradation of lignocellulose in vermicompost needs to happen in a nitrogenous environment such as in a sludge and animal waste matrix. E. fetida has been shown to decrease the cellulose content in paper mill sludge (Negi & Suthar, 2018). Our correlation analysis showed that the content of cellulose was negatively correlated with cellulase and the biomass of earthworms, which confirmed earthworms ingested and degraded cellulose on 28th and 56th day (Table 3). The swallowing and digesting of the worm were affected by SMS properties like the size of the parts. The initial SMS feedstock used in our study was a mixture of SMS and pig manure, which was composed of a variety of particles sizes. SMS mainly contains large particles, such as parts of corn cobs and sawdust. Pig manure is composed of many highly digested small particles. The addition of pig manure enhanced the proportion of small particles in SMS, which is beneficial for earthworm feeding. However, Pig manure is rich in ammonia and adding too much pig manure to SMS feedstock with its excessive amine content can be toxic to earthworms. Reasonable SMS feedstock should enhance the biomass of worms and promote SMS degradation. The addition of pig manure provided N nutrient for microbial growth and enhanced SMS degradation by microorganisms. Our results indicated that the suitable C:N ratio of initial feedstock in SMS vermicomposting was 20 or 25.

Table 3 Correlation coefficients of lignin content, cellulose content, cellulase and PPO activities in treatments with different C/N ratios of initial feedstocks.

	C/N ratio	Lignin	Cellulose	Cellulase	PPO	Biomass of earthworm	
28th day							
C/N ratio	1						
lignin	0.101	1					
cellulose	0.504	0.408	1				
cellulase	−0.595*	0.027	−0.698**	1			
PPO	0.266	−0.414	−0.230	0.294	1		
biomass of earthworm	−0.266	−0.283	−0.645**	0.739**	0.785**	1	
56th day							
C/N ratio	1						
lignin	0.324	1					
cellulose	0.529*	0.761**	1				
cellulase	0.084	−0.422	−0.626*	1			
PPO	0.342	−0.090	−0.258	0.594*	1		
biomass of earthworm	−0.098	−0.665**	−0.657**	0.797**	0.701**	1	
Notes.

* Correlation is significant at the 0.05 level.

** Correlation is significant at the 0.01 level.

Changes of biological characteristics in SMS vermicomposting

The biomass of earthworms generally increased within 28 days and decreased at 56 days (Table 2). Earthworm biomass was affected by the addition of pig manure. Within 28 days, the earthworm biomass increased in the S1, S2, and S3 treatments if the nutrient supply was sufficient, which is similar to the results of Gong et al. (2019). From 28 to 56 days, the decrease of earthworm biomass indicated nutrients depletion. The lowest mortality and highest biomass of earthworms in the S2 treatment demonstrated that it provided the best nutrient supply among all treatments. The meant S2 treatment provided sustainable conditions for SMS vermicomposting.

The production, stimulation, and inhibition factors of cellulase regulated vermicomposting activity. (1) Earthworms and microbes produce cellulose and the physiological conditions of earthworms and microorganisms can affect their secretion of cellulose. A proper C:N ratio of initial feedstock aids in growth and lifecycle of worms and microbes, which increase the activities of cellulase in vermicomposting (Aira, Monroy & Domínguez, 2006). The addition of pig manure can alter the C:N ratio of initial SMS feedstock. In this study, SMS feedstock supply determined the nutrients content for earthworms and microbes. The gradients of pig manure added to SMS resulted in variations of cellulose. There was a positive correlation between earthworm biomass and cellulase. The biomass of earthworms was positively correlated with the activities of cellulase and PPO on 28th and 56th day (Table 3), which indicated that the higher biomass of earthworms could increase the activities of cellulase and PPO. Similarly, Gong et al. (2019) reported that vermicompost enhanced the activities of cellulase, urease, and alkaline phosphatase. Our results verified that a greater biomass of earthworms would secrete more cellulase. (2) There were some stimulation factors affecting cellulase activity in vermicomposting. In this study, EC was increased by vermicompost in each treatment (Table 1), which meant that vermicomposting may enhance the fluidity of some metal ions. Previous reports showed that metal ions such as Ca2+, Mg2+, Mn2+, Zn2+, Pb2+, Fe2+, and Fe2+ had a positive influence on cellulase activity (Chen et al., 2019; Pachauri et al., 2018). The EC values and cellulase showed the same fluctuating trends as the C:N ratio of the initial feedstock. The higher EC values may illustrate the greater cellulase activities in the S2 treatment on 28, 42, and 56 days. (3) There were some inhibitory factors affecting cellulase activity in vermicomposting. Two lignin preparations can suppressed the six cellulases and one recombinant -1,4-endoglucanase (Berlin et al., 2006). Lignin contains three aromatic alcohols (coniferyl alcohol, sinapyl alcohol, and p-coumaryl alcohol) (Calvo-Flores & Dobado, 2010; Jiang, Nowakowski & Bridgwater, 2010; Menon & Rao, 2012). The aromatic substances can destroy the structure of cellulase and reduce its effects on vermicomposting. The removal of compounds with aromatic rings can effectively protect cellulase activity. PPO converted part of the aromatic compounds into quinones, which protected cellulase. During vermicomposting, the PPO activities were higher in the S2 treatment than the other treatments. Most of the phenols and the sesquiterpene lactones in parthenium were degraded by vermicomposting (Hussain, Abbasi & Abbasi, 2016). The protection of PPO may have resulted in the high cellulase activity in the S2 treatment.

The GI values of the vermicomposting products were higher than 100%, following the judgment standard of McLachlan, Chong & Vorony (2004). The vermicomposting products of SMS were non-phytotoxic.

Conclusions

The addition of pig manure promoted the fragmentation of SMS, enhanced the decomposition of lignocellulose, and increased the activities of cellulase and PPO in the SMS vermicomposting system. Physiochemically, the fragmentation of SMS and decomposing of lignocellulose increased with the addition of pig manure in the S0, S1, and S2 treatments, and decreased with the increased addition of pig manure in the S3 and S4 treatments. Biologically, the survival and biomass of earthworms were higher in the S2 group than the others. The activities of cellulase and PPO were at a high level in S2 treatment. It indicated that the changes of physical, chemical, and biological characteristics in SMS vermicomposting are directly related to the initial SMS feedstocks with different addition of pig manure, and it suggested that the optimal vermicomposting condition was the C/N ratio of initial feedstock at 25.

Supplemental Information

Supplemental Information 1 Physico-chemical characteristics of the various substrates used for vermicomposting

Values are mean ±  standard error (n = 3). Means in a row followed by different letters are significantly different at p < 0.05 according to Tukey HSD test.

Click here for additional data file.

Supplemental Information 2 Biomass and mortality rates of the earthworm (Eisenia fetida)

Values are mean ±  standard error (n = 3). Means in a row followed by different letters are significantly different at p < 0.05 according to Tukey HSD test.

Click here for additional data file.

Supplemental Information 3 Contents of cellulose (A) and lignin (B); degradation rate of cellulose (C) and lignin (D); and the activities of cellulase (E) and PPO (F) in different treatments during 56 days of vermicomposting

Vertical bars in the figures represent standard error of the means (n = 3).

Click here for additional data file.

Supplemental Information 4 Germination Indexes (GI) of Zea mays linn. Spp. (%)

1.25% extractions of SMS products in different treatments after 56 days of vermicomposting were used in this germination test. Vertical bars in the figures represent standard error of the means (n = 3).

Click here for additional data file.

Additional Information and Declarations

Competing Interests

Author Contributions

Data Availability

The authors declare there are no competing interests.

Yajing Shi, Zhenyu Wang and Yurong Wang conceived and designed the experiments, performed the experiments, analyzed the data, prepared figures and/or tables, authored or reviewed drafts of the paper, and approved the final draft.

The following information was supplied regarding data availability:

The raw measurements are available in the Supplemental Files.

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
