# Peer review of "Optimizing the amount of pig manure in the vermicomposting of spent mushroom (Lentinula) substrate"

_PeerJ, doi:10.7717/peerj.10584_

## Round 0.1 · original submission · Major Revisions

Dear Dr. Wang

The manuscript (ID 52176), entitled "Optimizing the amount of pig manure in the vermicomposting of spent mushroom (Lentinula) substrate )", was reviewed by experts in the field, and the reviewers have raised some issues. Your manuscript needs to be significantly improved according to the reviewer's comments before it is processed further.

Reviewer 1 ·

Basic reporting

Language is perfect throughout the manuscript.
References should need to include.
Article structure and tables are clear but need correction in figures.

Experimental design

Modification should be needed

Validity of the findings

Modification should be needed

Additional comments

Line number 90-91: include the relevant references and discuss about the issues of other studies.
Line number 107 – 109: include the analytical method
Lin2 112 – 116 – Experimental design is confusing. Did you analyzed the C/N ratio after mixing SMS with pig manure? Or the mixing ratio is based on separately measured C/N ratio of CMS and pig manure? How the C/N ratio reduced with increasing volume of pig manure or decreasing volume of CMS?
Line 118 – 121: Juvenile or adult worms? Worm used with empty gut content? Include the average weight per worm. How to maintain 60-80% humidity, any temperature control chamber used? Add all details.
Figure 1 and 2: Figure doesn’t mean anything without pointing. Physical characteristics means solid particles? Is that dark indicating the solid particles figure 1? Point with arrow mark in images. All images in figure 2 (A-E) is 5 µm size except 2F which is 10 µm, why?
Line 279 – 280: statement standing alone is meaningless, combined with previous line (line 277 to 280).
Correlation (ratio of feed mixture, worm growth and other compost parameters) study should be need for getting more clear conclusive evidence of the results.

·

Basic reporting

The manuscript deals with the study of optimizing the ratio of pig manure addition in SMS vermicomposting based on the comprehensive consideration of physical, chemical, and biological parameters.
According to the authors, the S2 treatment showed optimal processing in SMS vermicomposting and pig manure addition through various factors which includes suitable C/N ratio, higher PPO activities and degradation rate of lignin.

There are a couple of small details that I consider should be addressed before publication:
In line 203 --> check the units.
Fig 1 and 2 --> mention the images either in lowercase or uppercase.

Experimental design

No comment.

Validity of the findings

In fig 3 (f) you have shown the graphical representation of PPO activity. Can you explain why the S4 treatment shown a drastic decline from 42nd to 56th day.

Additional comments

Each part of the submitted manuscript: the introduction, materials and methods, results and discussion, and the conclusion are well described. The work is well designed and is well performed. The experimental results and the conclusions are pertinent and supported by the results.
However, the English should be checked again.

---

## Round 0.2 · accepted · Accept

Reviewers in the field have evaluated your revised manuscript. Based on the reviewer’s comments, your manuscript was significantly improved. So, I am pleased to inform you that your manuscript is accepted for publication in PeerJ.

Reviewer 1 ·

Basic reporting

No comment

Experimental design

Satisfactory

Validity of the findings

Explained well

Additional comments

There is no further change required for this manuscript.

·

Basic reporting

Language is thorough throughout the manuscript.
The corrections made in the manuscript was content.

Experimental design

No comment.

Validity of the findings

The corrections made in the manuscript was content and clear.